# The Dietary Inflammatory Index and Chronic Lymphocytic Leukaemia in the MCC Spain Study

**DOI:** 10.3390/nu12010048

**Published:** 2019-12-23

**Authors:** José Carlos Flores, Esther Gracia-Lavedan, Yolanda Benavente, Pilar Amiano, Dora Romaguera, Laura Costas, Claudia Robles, Eva Gonzalez-Barca, Esmeralda de la Banda, Esther Alonso, Marta Aymerich, Elias Campo, Trinidad Dierssen-Sotos, Rafael Marcos-Gragera, Marta María Rodriguez-Suarez, Marta Solans, Eva Gimeno, Paloma Garcia Martin, Nuria Aragones, Nitin Shivappa, James R. Hébert, Marina Pollan, Manolis Kogevinas, Silvia de Sanjose, Gemma Castaño-Vinyals, Delphine Casabonne

**Affiliations:** 1Departament de Ciències Experimentals i de la Salut, Universitat Pompeu Fabra, Barcelona 08002, Spain; josecarlosfl@outlook.com (J.C.F.); esther.gracia@isglobal.org (E.G.-L.); manolis.kogevinas@isglobal.org (M.K.); 2Consortium for Biomedical Research in Epidemiology and Public Health (CIBERESP), Madrid 28029, Spain; ybenavente@iconcologia.net (Y.B.); epicss-san@euskadi.eus (P.A.); trinidad.dierssen@unican.es (T.D.-S.); rafael.marcos@udg.edu (R.M.-G.); martasolans@gmail.com (M.S.); nuria.aragones@salud.madrid.org (N.A.); mpollan@isciii.es (M.P.); sdesanjose@path.org (S.d.S.); 3Instituto de Salud Global de Barcelona (ISGlobal), Barcelona 08003, Spain; dora.romaguera@isglobal.org; 4Unit of Molecular Epidemiology and Genetic in Infections and Cancer (UNIC-Molecular), Cancer Epidemiology Research Programme (IDIBELL), Catalan Institute of Oncology, L’Hospitalet de Llobregat 08908, Spain; lcostas@iconcologia.net; 5Public Health Division of Gipuzkoa, BioDonostia Research Institute, San Sebastian 20014, Spain; 6Instituto de Investigación Sanitaria Illes Balears (IdISBa), Palma 07120, Spain; 7CIBER Fisiopatología de la Obesidad y Nutrición (CIBEROBN), Madrid 28029, Spain; 8Unit of Information and Interventions in Infections and Cancer (UNIC-I&I), Cancer Epidemiology Research Programme, (IDIBELL), Catalan Institute of Oncology, L’Hospitalet de Llobregat 08908, Spain; crobles@idibell.cat; 9Haematology, Bellvitge Biomedical Research Institute (IDIBELL), Catalan Institute of Oncology, L’Hospitalet de Llobregat 08908, Spain; e.gonzalez@iconcologia.net; 10Haematology Laboratory, Department of Pathology, Hospital Universitari de Bellvitge, L’Hospitalet de Llobregat 08908, Spain; edelabanda@bellvitgehospital.cat (E.d.l.B.); ealonso@bellvitgehospital.cat (E.A.); 11Hematopathology Unit, Department of Pathology, Hospital Clínic, (IDIBAPS), Barcelona 08036, Spain; aymerich@clinic.cat (M.A.); ECAMPO@clinic.cat (E.C.); 12Centro de Investigación Biomédica en Red Cáncer (CIBERONC), Instituto de Salud Carlos III (ISCIII), Madrid 28029, Spain; 13Marqués de Valdecilla Research Institute (IDIVAL), University of Cantabria, Santander 39011, Spain; 14Research group on Statistics, Econometrics and Health (GRECS), University of Girona, Girona 17071, Spain; 15Epidemiology Unit and Girona Cancer Registry, Catalan Institute of Oncology, Girona 17007, Spain; 16Universidad de Oviedo, área de medicina Preventiva y Salud Pública, Oviedo 33003, Spain; mrstsf@gmail.com; 17Hospital Universitario Central de Asturias (HUCA), Oviedo 33011, Spain; 18IUOPA: Instituto de Oncología de Asturias (IUOPA), Oviedo 33003, Spain; 19Haematology Department, Hospital del Mar, Barcelona 08003, Spain; 94015@parcdesalutmar.cat; 20Unidad de Gestión Clínica de Hematología. Hospital Universitario San Cecilio PTS de Granada, Granada 18016, Spain; paloma.garcia.martin.sspa@juntadeandalucia.es; 21Epidemiology Section, Public Health Division, Department of Health of Madrid, Madrid 28035, Spain; 22Cancer Prevention and Control Program, University of South Carolina, Columbia, SC 29208, USA; shivappa@mailbox.sc.edu (N.S.); jhebert@mailbox.sc.edu (J.R.H.); 23Department of Epidemiology and Biostatistics, Arnold School of Public Health, University of South Carolina, Columbia, SC 29208, USA; 24Department of Nutrition, Connecting Health Innovations LLC (CHI), Columbia, SC 29201, USA; 25Cancer Epidemiology Unit, National Centre for Epidemiology, Carlos III Institute of Health, Madrid 28029, Spain; 26Hospital del Mar Medical Research Institute (IMIM), Barcelona 08003, Spain; 27PATH, Sexual and Reproductive Health, Seattle, WA 98121, USA

**Keywords:** dietary inflammatory index, chronic lymphocytic leukaemia, case-control study, MCC Spain study, nutrition, cancer

## Abstract

Chronic inflammation plays a role in the development of chronic lymphocytic leukaemia (CLL), and diet might modulate chronic inflammation. This study aims to evaluate the association between the dietary inflammatory index (DII^®^) and CLL. A total of 366 CLL cases and 1643 controls of the Spanish multicase-control (MCC) Spain study were included. The inflammatory potential of the diet was assessed using the energy-adjusted dietary inflammatory index (E-DII) based on 30 items from a validated semi-quantitative food frequency questionnaire. Odds ratios (OR) and 95% confidence intervals (CI) were estimated using logistic regression models controlling for potential confounders. Overall, a modest, non-statistically significant, positive association was observed between CLL and E-DII scores (OR for a one-unit increase in E-DII: 1.05 (CI 95%: 0.99, 1.12), *p*-value = 0.09 and by tertiles: OR_T2vsT1_: 1.20 (CI 95%: 0.90, 1.59); OR _T3vsT1_: 1.21 (CI 95%: 0.90, 1.62), *p* trend = 0.21). These results were independent from disease severity (*p-*het: 0.70), time from diagnosis (*p-*het: 0.67) and CLL treatment received (*p-*het: 0.56). No interactions were detected. In conclusion, the consumption of a diet with high pro-inflammatory components was not significantly associated with CLL. Changes towards a more pro-inflammatory dietary pattern in younger generations not included here warrant future research.

## 1. Introduction

Chronic lymphocytic leukaemia (CLL) is the most common form of leukaemia in the Western world [1]. The aetiology of CLL is still unclear. It is known, however, that the risk of developing CLL increases with age and among those with a first-degree relative with haematological cancers [2]. Men have approximately twice the risk of developing CLL than women [3], and incidence rates are higher amongst Whites than Blacks or Asians [4]. A pooled analysis of 2440 CLL cases and approximately 15,186 controls from 13 case-control studies within the InterLymph consortium showed that an increased risk of developing CLL was associated with having a family history of haematological cancers, being taller and having worked in a farm. On the other hand, exposure to UV radiation and atopic conditions were inversely associated with CLL [5]. Regarding diet, case-control studies have obtained controversial results for individual nutrients such as meat, dairy products and vegetables [6,7,8,9,10,11,12,13,14,15,16,17].

Inflammation is a normal biological process and acute inflammatory responses are necessary for mounting a competent immune response [18]. However, chronic low-grade inflammation may result from a variety of environmental insults; and it is particularly common among obese people [19,20]. Chronic, low-grade systemic inflammation has been associated with an increased risk of chronic conditions such as cardiovascular disease and cancer [21,22,23]. Autoimmune and chronic inflammatory diseases seem to potentiate genetic events and contribute to B-cell lymphoid malignancies, which result in different subtypes of lymphoma [24]. In particular, chronic inflammation contributes to the pathobiology and symptomatology of CLL as supported by a comprehensive analysis of 23 cytokines, which found that the levels of 17, mostly pro-inflammatory, cytokines were significantly higher in the sera of CLL patients than in the sera of healthy individuals [25]. A large body of evidence supports that diet plays a fundamental role in the regulation of chronic inflammation through both pro- and anti-inflammatory mechanisms [26,27,28,29,30,31,32].

In 2009, Cavicchia PP et al. reported on the first version of the dietary inflammatory index (DII^®^) [33]. The DII is a tool designed to measure the inflammatory potential of a person’s diet and categorise it on a continuum from maximally anti- to maximally pro-inflammatory. The DII is based on the results of an extensive literature search on the association between various inflammatory biomarkers and dietary components [33,34]. Many studies around the world have been carried out aiming to find an association between DII and cancer outcomes. A meta-analysis, that used 24 studies (13 case-control studies, six prospective cohorts, one retrospective cohort, three randomised control trials (RCTs) and one unspecified study design) [35] found strong positive and significant associations between a pro-inflammatory diet (measured by a higher DII) and cancer incidence and mortality across cancer types, study populations and study design. Data on CLL are sparse and, to our knowledge, this type of association has only recently been examined in the European Prospective Investigation into Cancer and Nutrition (EPIC). This study, including 2606 non-Hodgkin lymphoma (NHL) cases and 537 CLL cases, used a modified version of the DII and showed no association between the inflammatory score of the diet and overall lymphoma as well as CLL risk [36]. In a couple of Italian case-control studies, DII has been shown to be associated with Non-Hodgkin’s Lymphoma but not with Hodgkin’s Lymphoma [37,38]. 

Hence, by using data from the Spanish multi case-control (MCC) Spain study, the present investigation aims to analyse the association between the DII and CLL. We hypothesised that people with a more pro-inflammatory diet (measured by a higher DII score) are more likely to develop CLL.

## 2. Materials and Methods 

### 2.1. Study Population

This study uses CLL cases and controls from the MCC Spain study. The MCC Spain study is a population-based multi case-control study carried out between September 2008 and December 2013 in 23 hospitals in 12 Spanish provinces (Asturias, Barcelona, Cantabria, Girona, Granada, Gipuzkoa, Huelva, León, Madrid, Murcia, Navarra and Valencia, list S1). This study aims to evaluate the influence of environmental exposures and their interaction with genetic factors in some of the most common tumours in Spain. CLL was incorporated later to the study as part of a collaboration with the International Cancer Genome Consortium [39]. Controls were randomly selected from the administrative records of selected primary care health centres located within these hospitals’ catchment area and invited to the study through telephone. Cases and controls were recruited simultaneously, and population-based controls were frequency-matched to cases according to the province of recruitment sex and age. The response (agreement to enrol) rate for CLL cases was 87%, while the overall mean participation rate of controls was 53% and varied by region. Here, subjects lacking information on diet (used to calculate the DII) and subjects with information on energy (kcal/day) in percentile 1 and below and percentile 99 and above were excluded. Due to a very small sample size, non-White subjects were also excluded. A total of 1643 CLL controls and 366 CLL cases were included for this study (Figure 1). More information on the MCC Spain study subjects and overall study design can be found elsewhere [39].

### 2.2. Outcome Definition

CLL cases were diagnosed following the International Workshop on CLL criteria: presence of ≥5 × 10^9^ monoclonal B-cell lymphocytes/l (characterised by the co-expression of CD5 antigen and B-cell surface antigens CD19, CD20 and CD23) [1]. Chronic lymphocytic leukaemia (CLL) and small lymphocytic lymphoma considered to be the same underlying disease were labelled as CLL [1]. Cases were histologically confirmed and, as CLL follows an indolent course, those interviewed ≥1 year from diagnosis were also recruited [39]. Disease severity was assessed using the Rai staging system and then categorised into two groups, low-risk category (Rai 0) and intermediate/high risk category (Rai I–IV) [40].

### 2.3. Data Collection and Sources

For the MCC Spain study, data on sociodemographic factors, lifestyle and personal/family history were collected during face-to-face interviews. A semi-quantitative Food Frequency Questionnaire (FFQ) was used in order to examine diet. The FFQ was a modified version of a previously validated tool in Spain [41]. The questionnaire was modified in order to include regionally available food products. It included a total of 140 food items and evaluated the subjects’ typical dietary intake during the preceding year. For each item, portion sizes were specified, and photographs were used in order to determine degrees of cooking. Data from this FFQ were used for this study in order to assess the inflammatory potential of the subjects’ diet. The questionnaire was self-administrated and returned by mail, although in some instances it was completed face-to-face. Data on important dietary pattern changes in the previous 5 years and on vitamin and mineral supplement intake were also gathered [39]. The global response rate of the FFQ was 88%. A copy of the FFQ can be found here: http://www.mccspain.org.

### 2.4. Dietary Inflammatory Index

The energy-adjusted DII^®^ (E-DII) calculation was carried out using the 2014 adapted version of Shivappa et al.’s DII [34]. The DII is based on the results of an extensive literature search from 1950 to 2010 on inflammatory biomarkers (IL-1β, IL-4, IL-6, IL-10, TNFα and CRP) and diet [33]. This tool has been validated by analysing serum high-sensitivity C-reactive protein, a biomarker of a low level of inflammation, in the SEASONS study [33]. Here, 30 out of the 45 original food parameters of the DII score (alcohol, vitamin B12, vitamin B6, carbohydrate, cholesterol, total fat, fibre, folic acid, garlic, iron, magnesium, monounsaturated fatty acids, niacin, onion, protein, polyunsaturated fatty acids, riboflavin, saturated fat, thiamin, vitamin A, vitamin C, vitamin D, vitamin E, zinc, flavan-3-ol, flavones, flavonols, flavonones, anthocyanidins, and isoflavones) were available from the MCC Spain study diet database used. Computation of the E-DII scores relied on an energy-adjusted global database and the scores were calculated by converting raw dietary components to amount per 1000 kcal. The higher the E-DII score, the more pro-inflammatory the diet. More information on the construction of the DII can be found elsewhere [34].

### 2.5. Statistical Analyses

For the descriptive analyses, Student t-tests and chi-square tests were used for continuous and categorical variables, respectively, in order to assess differences between cases and controls. Associations between the E-DII (as a continuous variable and categorised in tertiles based on controls distribution) and CLL were evaluated using binary and multinomial logistic regression models. Trend tests using E-DII tertiles categorised as 1, 2 and 3 as well as using the E-DII means and medians within each E-DII tertile were examined. Basic models were adjusted for sex, age (continuous), education (primary, secondary, university) and province of residence (Barcelona vs. other regions). Further variables were examined individually as potential confounders: body mass index (BMI) (continuous); weight (continuous); height (continuous); sex-specific waist-to-hip ratio for higher metabolic complication risk (≥0.90 men, ≥0.85 women [42]); alcohol consumption at time of interview (continuous); smoking (never, current, former); first-degree family history of haematological cancer (yes/no); type II diabetes (yes/no); total energy intake (continuous); physical activity (in the last 10 years, measured in Metabolic Equivalent of Task (METs)/week: inactive (0) low (0.1–8), moderate (8–15.9) and very active (≥16)); and ever worked in agriculture (yes/no). Missing values for controls/cases, respectively, were: weight (22 (1.3%)/1 (0.27%)); height (52 (3.2%)/12 (3.3%)); waist-to-hip ratio (14 (0.9%)/2 (0.6%)); diabetes (21 (1.3%)/6 (1.6%)); smoking (5 (0.3%)/2 (0.6%)); physical activity (38 (2.3%), 13 (3.6%)); ever worked in agriculture (3 (0.2%)/1 (0.3%)); first-degree family history haematological cancers (108 (6.6%)/15 (4%)); treatment 3 CLL (0.8%); disease severity 9 CLL (2.5%). A generalised additive model (GAM) was carried out in order to examine the linearity of the association between E-DII and CLL. When evaluating the potential effect of confounders, the basic model was contrasted with the complete case analysis for each possible confounder variable. Following the use of a Directed Acyclic Graph (DAG) (Appendix A), models were further adjusted by height (continuous), smoking (never, current, former,) and ever worked in agriculture (yes, no). However, these variables were not included in the final models alone or in combination because they did not alter the overall estimates (% change in OR for a one-unit increase <2%) (Appendix A). As part of a sensitivity analysis, tests for heterogeneity by disease severity (measured by the Rai Staging System), time from diagnosis and CLL treatment received were performed. Both multiplicative and additive interaction analyses between pre-selected variables (age, sex, alcohol consumption, height, smoking, agriculture) and E-DII (continuous) were also carried out using log-likelihood ratio tests, relative excess risk due to interaction, attributable proportion and synergy index.

Data analyses were carried out using STATA 14 (StataCorp. 2015. Stata Statistical Software: Release 14. College Station, TX: StataCorp LP, USA) with a two-sided statistical significance set at *p* < 0.05.

## 3. Results

### 3.1. Characteristics of Cases and Controls in the MCC Spain Study

The distribution of CLL cases and controls according to the selected variables is given in Table 1. CLL cases were slightly older, tended to consume less alcohol (g/day), had a higher waist-to-hip ratio, had a stronger first-degree family history of haematological cancers and were more likely to have worked in agriculture. No other statistically significant differences between cases and controls were identified for the other analysed variables. The median value of the E-DII was −0.44 and its range varied between −5.64 and 5.47. The distribution of key characteristics of controls according to their crude E-DII means is shown in Appendix A. Controls with a higher E-DII score were more likely to be younger and male; consumed more calories (kcal/day); were taller and current smokers; had a lower education level; did less physical activity; had never worked in agriculture; and had lower rates of type II diabetes.

### 3.2. E-DII and CLL Risk

A modest, non-statistically significant, positive association between E-DII (continuous) and CLL was found (OR for a one-unit increase in E-DII: 1.05 (CI 95%: 0.99, 1.12), *p*-value = 0.09). A non-statistically significant positive association was observed when evaluating the association with E-DII categorised in tertiles OR _T2vsT1_: 1.20 (CI 95%: 0.90, 1.59); OR _T3vsT1_: 1.21 (CI 95%: 0.90, 1.62), *p* trend = 0.21) (Table 2). There was no difference in trend tests following the use of categorical values (1, 2 and 3), means or medians for each E-DII tertile (data not shown). This association was also evaluated using sex-specific E-DII tertiles without materially changing the results (data not shown). The GAM did not provide support for a non-linear association (*p* = 0.18) (Figure 2). No statistically significant additive nor multiplicative interactions were detected (Appendix A).

### 3.3. Sensitivity Analyses.

Results of a sensitivity analysis showed no statistically significant heterogeneity by disease severity (OR_Rai 0_ = 1.05 (95% CI: 0.97, 1.13); OR _Rai I–IV_ = 1.06 (95% CI: 0.97, 1.16), *p*-value for heterogeneity = 0.70), time from diagnosis (OR_<1 year_ = 1.03 (95% CI: 0.93, 1.15); OR_1 to 3years_ = 1.06 (95% CI: 0.99, 1.14), *p*-value for heterogeneity = 0.67) or CLL treatment received (OR_untreated_ = 1.04 (95% CI: 0.98, 1.12); OR_treated_ = 1.08 (95% CI: 0.96, 1.22) *p-*value for heterogeneity = 0.56) (Table 3).

## 4. Discussion

This first case-control study on the association between the E-DII and CLL observed a non-statistically significant positive association between a pro-inflammatory diet, measured by the E-DII, and CLL. 

For years, nutritional epidemiological studies have focused on individual foods or nutrients. However, people eat a combination of nutrients, and therefore these analyses have several theoretical and methodological drawbacks [43]. On the one hand, prospective studies looking for associations between CLL and nutrients or individual food items have found positive associations with poultry and processed meat [44], total fat [45] or carbohydrate [46] intake (in women), and inverse associations with the consumption of isoflavones [47]. On the other hand, case-control studies have found inconsistent evidence for the associations between CLL and dairy products, fish, meat, fruits and vegetables [6,7,8,9,10,11,12,13,14,15,16,17]. Due to this, in recent decades, nutritional epidemiology has transitioned toward studying dietary patterns in order to assess their risk of disease [43]. Previously, diet has been identified as a chronic inflammation regulator [26,27,28,29,30,31,32] and, in turn, chronically elevated inflammation itself has been associated with increased risk of chronic conditions such as cardiovascular disease and cancer [21,22,23]. Thus, the relevance of using tools such as the E-DII as another way of evaluating dietary patterns and their relationship with the development of disease. The biological plausibility in the role of inflammation and the pathogenesis of lymphoma has been studied previously. Certain autoimmune and chronic inflammatory conditions, such as rheumatoid arthritis, have been identified as strong lymphoma risk factors [48,49,50]. Given that CLL comes from a lymphoid type cell, sharing many characteristics with lymphomas [1], an important role of inflammation in the pathogenesis of CLL is likely.

There is mounting evidence regarding the association between DII and solid neoplasms. Clear associations have been reported for colorectal cancers [51,52] and gastric cancers [53], whereas the association with breast cancer is still inconclusive [54,55,56]. Regarding haematological cancers, other case-control studies looking into the association of DII (and DII-like tools) and other types of haematological cancers have been carried out [37,38], reporting positive associations between a pro-inflammatory diet and NHL and B-cell lymphoma subtypes. These findings were supported by the European Prospective Investigation into Cancer and Nutrition (EPIC) study for NHL and all mature B-cell subtypes [36]. 

Regarding the effects of a pro-inflammatory diet on CLL development, to our knowledge, this association has only recently been examined in the EPIC study. This study, including 537 CLL cases, used a modified version of the E-DII and showed no association between the inflammatory score of the diet and CLL risk [36]. However, in the EPIC study, diet through FFQ was only measured at the beginning of the investigation in the 1990s. Therefore, changes in dietary patterns and exposure to inflammatory components might not have been properly captured. Altogether, results of the EPIC study and this MCC Spain study suggest that chronic inflammation associated with diet might not play a strong role in CLL aetiology. 

Incidence rates of CLL are significantly lower in Asia than in Western countries [57] and this incidence patterns tend to change with immigration and acculturation. For instance, an American study found out that rates of CLL were statistically significantly lower amongst foreign-born Asians than US-born Asians [58]. Likewise, a Taiwanese study suggests that the Westernisation of lifestyle in Taiwan since the 1960s could explain the drastic increase of incidence of CLL in the country’s population [59]. Although the much lower CLL incidence rates in Asians compared to Europeans/Whites support the importance of genetic background in the disease, these studies support the idea that the aetiology of this lymphoid malignancy subtype might involve environmental exposures that are more common in Western populations. Likewise, a Western-type diet, high in sugar intake, fried foods, high-fat dairy products and refined grains, has been associated with higher levels of inflammatory biomarkers, such as C-reactive protein (CRP) [26,27,28]. Interestingly, using the MCC Spain data, an association between CLL and a high adherence to a Western dietary pattern was reported [40], suggesting that factors other than inflammation associated with a Western diet should be evaluated in future studies.

The main strength of this study is that we were able to evaluate the effect of important clinical, sociodemographic and lifestyle variables when examining the association between E-DII and CLL. This information allowed the evaluation of many possible confounding variables and the performance of different sensitivity analyses to support our research. Additionally, given the almost universal coverage of the Spanish National Health System, selecting controls through lists of general practitioners, contributes to a representative sampling frame. Furthermore, in comparison to other E-DII and inflammation-focused studies we had important information on flavonoids, identified many times as anti-inflammatory agents [60], when calculating the score. 

Some limitations should also be considered when interpreting the results. First of all, as in any case-control study, selection and recall biases might have been present. By using a self-administered FFQ, measurement errors in the estimation of foods intake are likely. Nonetheless, the FFQ used in MCC Spain study is a validated tool and followed the methodology outlined in Calvert et al. [61] in order to adjust the responses of the FFQ. Inclusion of prevalent cases might also be of concern as the aetiology of the disease of people who have survived might be different than from those dying soon after diagnosis. By the same token, patients who have survived the disease might have changed. their diet considerably since they were diagnosed. However, the sensitivity analyses carried out suggested that the inclusion of prevalent cases might not have biased our results. When evaluating subgroups and interactions, we had less statistical power to detect significant associations due to relatively small sample size. We observed the strongest positive association between the E-DII and CLL in never smokers, whereas null associations were observed among former and current smokers. While the interaction was not statistically significant, this result suggests that the impact of the inflammation induced by the diet might be modulated by smoking or, alternatively, the effect of diet could be simply masked given the strong pro-inflammatory effect of smoking. Although the effect of confounders was evaluated using statistical methods and a DAG, residual confounding or unknown confounders related to the exposure cannot be totally dismissed. Finally, regarding the E-DII, we were limited to information of only 30 out of the original 45 food parameters. Furthermore, regarding the anti-inflammatory property given to alcohol, this has only been reported for low/moderate alcohol consumers (less than 30–40 g/day) in some studies [62,63]; hence, the E-DII (with a negative weight of −0.278 per gram of alcohol intake per day) might not completely capture the inflammatory potential of alcohol.

## 5. Conclusions

Our research suggested that a more pro-inflammatory diet, determined by a higher E-DII score, may not be associated with CLL. Furthermore, given that major changes in diet may not be captured in the two studies looking into this association, continuous evaluation of dramatic changes in dietary patterns and their impact on health is needed in order to confirm these findings.

## Figures and Tables

**Figure 1 nutrients-12-00048-f001:**
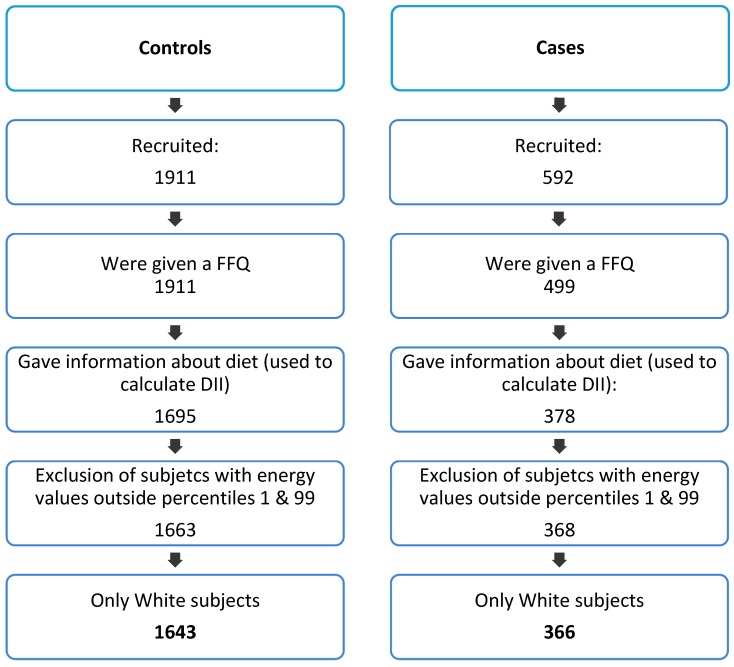
Flow chart of the population of the study.

**Figure 2 nutrients-12-00048-f002:**
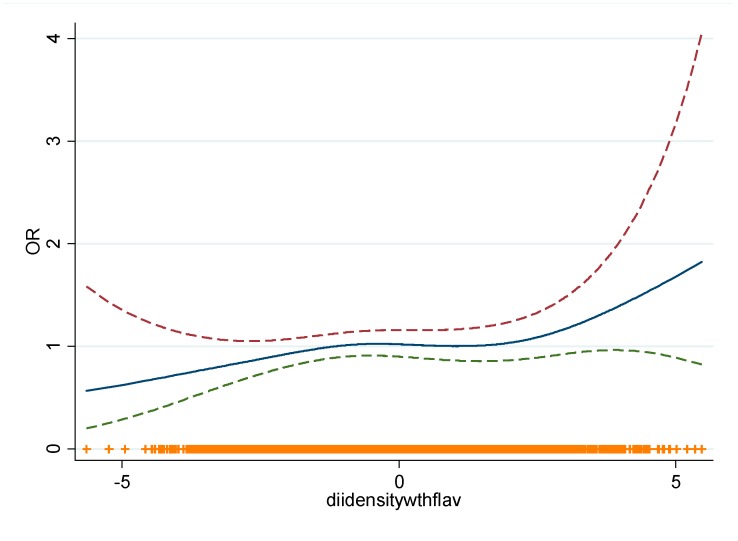
Generalised additive model of the association between E-DII and CLL (*p* = 0.18). Blue line: Spline; Dotted lines: 95% confidence interval. X-axis: E-DII

**Table 1 nutrients-12-00048-t001:** Baseline characteristics of chronic lymphocytic leukaemia (CLL) cases and controls in the multicase-control (MCC) Spain study.

Variables **	Controls	Cases	*p* Value ^a^
**E-DII, mean (SD)**	−0.20 (2.0)	−0.13 (2.0)	0.51
**Age (years), mean (SD)**	63.5 (11.4)	66.2 (10.2)	**<0.001**
**Male, N (%)**	939 (57.2)	215 (58.7)	0.58
**Education, N (%)**			0.81
Primary	861 (52.4)	197 (53.8)	
Secondary	481 (29.3)	107 (29.2)	
University	301 (18.3)	62 (16.9)	
**Region, N (%)**			**<0.001**
Barcelona	889 (54.1)	240 (65.6)	
Asturias	208 (12.7)	50 (13.7)	
Cantabria	324 (19.7)	21 (5.7)	
Granada	148 (9.0)	27 (7.4)	
Gerona	74 (4.5)	28 (7.7)	
**BMI (kg/m2), mean (SD)** ^b^	26.9 (4.5)	27.3 (4.4)	0.13
**Weight (kg), mean (SD)**	73.9 (13.7)	75.0 (14.3)	0.15
**Height (cm), mean (SD)**	165.7 (8.5)	166.0 (9.1)	0.61
**Waist-to-hip ratio, N (%)** with higher risk of metabolic complications ^c^	1239 (76.1)	317 (87.1)	**<0.001**
**Energy (kcal/day), mean (SD)**	1891.4 (573.3)	1934.3 (601.7)	0.20
**Alcohol consumption (gr/day), mean (SD)** ^d^	12.0 (17.1)	9.5 (13.8)	**0.008**
**Ever had diabetes, N (%) **	236 (14.6)	46 (12.8)	0.38
**Smoking, N (%)**			0.85
Never	722 (44.1)	164 (45.1)	
Current smoker	290 (17.7)	60 (16.5)	
Former smoker	626 (38.2)	140 (38.5)	
**Physical activity, N (%)** ^e^			0.62
Inactive	668 (41.6)	134 (38.0)	
Low	227 (14.1)	55 (15.6)	
Moderate	196 (12.2)	47 (13.3)	
Very active	514 (32.0)	117 (33.1)	
**Ever worked in agriculture, N (%)**	325 (19.8)	107 (29.3)	**<0.001**
**First-degree family history of haematological cancer, N (%)**	53 (3.5)	36 (10.3)	**<0.001**
**Treated for CLL, N (%)**	N/A	79 (21.8)	N/A
**Disease severity, N (%)** ^f^			N/A
Rai 0	N/A	207 (58.0)	
Rai I–IV	N/A	150 (42.0)	
**Time from diagnosis to recruitment**			N/A
<1 year	N/A	97 (26.5)	
≥1year	N/A	269 (73.5)	

** Missing values for controls/cases: weight (22 (1.3%)/1 (0.27%)); height (52 (3.2%)/12 (3.3%)); waist-to-hip ratio (14 (0.9%)/2 (0.6%)); diabetes (21 (1.3%)/6 (1.6%)); smoking (5 (0.3%)/2(0.6%)); physical activity (38 (2.3%), 13 (3.6%)); ever worked in agriculture (3 (0.2%)/1 (0.3%)); first-degree family history haematological cancer (108 (6.6%)/15 (4%)); treated for CLL 3 (0.8%); disease severity 9 CLL (2.5%). N: total number, SD: standard deviation, BMI: body mass index, E-DII: energy-adjusted dietary inflammatory index. In bold *p* ≤ 0.05; N/A: non-applicable. a: *P*-values for heterogeneity calculated with the Student t-test for continuous variables and with Chi-square test for categorical variables. b: BMI variable obtained through a basic imputation method. c: Waist-to-hip ratio risk categories according to WHO criteria [42]. d: Alcohol consumption at time of interview. e: Physical activity in the last 10 years measured METs/week: inactive (0), low (0.1–8), moderate (8–15.9), and very active (≥16). f: Measured by the Rai Staging System.

**Table 2 nutrients-12-00048-t002:** Odds ratios and 95% confidence intervals of CLL according to E-DII among 1643 CLL controls and 366 CLL cases in the MCC Spain study.

	By Tertiles of E-DII			*P*-value Trend	One-Unit Increase in E-DII	*P*-value Trend
T(min, max)	T1 (−5.64, −1.31)	T2 (−1.31, 0.59)	T3 (0.59, 5.47)			
N controls/cases	547/114	549/131	547/121		1643/366	
OR ^a^ and 95% CI	Ref	1.20 (0.90, 1.59)	1.21 (0.90, 1.62)	0.21 ^c^	1.05 (0.99, 1.12)	0.09
N controls/cases					1586/352	
OR ^b^ and 95% CI	Ref	1.24 (0.93, 1.66)	1.24 (0.91, 1.68)	0.17 ^c^	1.06 (1.00, 1.13)	0.07

E-DII: energy-adjusted dietary inflammatory index; OR: odds ratio; 95% CI: 95% confidence interval; T: tertiles based on control distribution; Ref: reference category; DAG: Directed Acyclic Graph; min: minimum; max: maximum; N: number; ^a^ Basic model is E-DII adjusted for sex, age, education (primary, secondary, university) and region (Barcelona region vs. other regions); ^b^ Basic model is further adjusted for: height (continuous); smoking (never, former, current); ever worked in agriculture; based on DAG results; ^c^
*p* trend for linear trend test (categories 1, 2, 3).

**Table 3 nutrients-12-00048-t003:** Association between the energy-adjusted dietary inflammatory index and chronic lymphocytic leukaemia by severity of disease, time from diagnosis to recruitment and CLL treatment in the multicase-control (MCC) Spain study.

	E-DII				One-Unit Increase in E-DII		
	OR ^a^ and 95% CI				OR ^a^ and 95% CI		
	T1	T2	T3	*p-*Value for Trend ^b^		*p-*Value	*p-*Value for Heterogeneity ^c^
**Rai stages**							
N controls/C1/C2	547/69/43	549/66/61	547/72/46				
C1: Rai 0	Ref	1.04 (0.72, 1.50)	1.30 (0.90, 1.89)	0.17	1.05 (0.97, 1.13)	0.27	
C2: Rai I–IV	Ref	1.42 (0.94, 2.14)	1.09 (0.69, 1.72)	0.70	1.06 (0.97, 1.16)	0.17	0.70
**Time from diagnosis to recruitment**							
N controls/C1/C2	547/30/84	549/36/95	547/31/90				
C1: <1 year	Ref	1.25 (0.75, 2.07)	1.14 (0.66, 1.95)	0.63	1.03 (0.93, 1.15)	0.55	
C2: ≥1 year	Ref	1.18 (0.86, 1.63)	1.23 (0.88, 1.72)	0.22	1.06 (0.99, 1.14)	0.09	0.67
1–3 years	Ref	0.93 (0.60 to 1.47)	1.14 (0.72 to 1.78)	0.58	1.03 (0.94 to 1.14)	0.51	
≥3 years	Ref	1.45 (0.95 to 2.22)	1.33 (0.84 to 2.12)	0.21	1.09 (0.99 to 1.20)	0.07	
**Treated for CLL**							
N controls/C1/C2	547/95/19	549/94/35	547/95/25				
C1: No	Ref	1.05 (0.76, 1.43)	1.18 (0.86, 1.64)	0.31	1.04 (0.97, 1.12)	0.22	
C2: Yes	Ref	1.80 (1.01, 3.20)	1.30 (0.69, 2.44)	0.46	1.08 (0.96, 1.22)	0.20	0.56

E-DII: energy-adjusted dietary inflammatory index; OR: Odds Ratio; 95% CI: 95% Confidence Interval; C: Category; Ref: reference category. ^a^ Basic model adjusted for sex, age (continuous), education (primary, secondary, university) and region (Barcelona region vs. other regions). ^b^
*p* trend for linear trend test (assigning the E-DII mean value of each tertile and testing as an ordinal value in the model). ^c^
*p* value for the heterogeneity of effects.

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
