# Peer review of "The Dietary Inflammatory Index and Chronic Lymphocytic Leukaemia in the MCC Spain Study"

_nutrients, 2019, doi:10.3390/nu12010048_

Round 1

Reviewer 1 Report

José Carlos Flores et al describe in "THE DIETARY INFLAMMATORY INDEX AND CHRONIC LYMPHOCYTIC LEUKAEMIA IN THE MCC-SPAIN STUDY" the effects of diet preferences in a Spanish population and the incidence of CLL. 

The study is well designed, and presents the results clearly, despite the fact that the authors did not find any significant association between diet and incidence of CLL.

Minor comments:

On tables 2 and 3 several entrances have 'REF", please specify the meaning or add a reference.

Author Response

Reviewer #1:

The study is well designed, and presents the results clearly, despite the fact that the authors did not find any significant association between diet and incidence of CLL.

Minor comments:

On tables 2 and 3 several entrances have 'REF", please specify the meaning or add a reference.

Answer:

We thank the reviewer for her/his comments. We have added the definition of “ref” in the footnote of Tables 2 and 3.

Reviewer 2 Report

The study analyzed the association between the dietary inflammatory index and CLL in the MCC-SPAIN study. The study is well conducted and provided information that consumption of more pro-inflammatory diets with higher E-DII scores may not be associated with CLL. The statistical analyses of this study were fairly conducted and the conclusion was supported by the results. The reviewer considered the manuscript fits the general interests of the audience of the journal.

Author Response

Reviewer #2:

Comments and Suggestions for Authors

The study analyzed the association between the dietary inflammatory index and CLL in the MCC-SPAIN study. The study is well conducted and provided information that consumption of more pro-inflammatory diets with higher E-DII scores may not be associated with CLL. The statistical analyses of this study were fairly conducted and the conclusion was supported by the results. The reviewer considered the manuscript fits the general interests of the audience of the journal.

Answer:

We thank the reviewer for her/his comments.

Reviewer 3 Report

This is a well-conducted case control study of chronic lymphocytic leukemia in Spain.  The well thought out logistic analysis showed no statistical association with the Dietary Inflammatory Index.  Here are a few suggestions for changes.

Abstract

Line 74.  I suggest adding the word “significantly” before “associated”.  This will stress that while the DII-CLL association did not meet the p<0.05 threshold, there was a clear trend in that direction.

Line 75. I suggest removing the word “Dramatic” here.  Given the modest findings of this study, it is unlikely that dramatic findings in a younger population can be expected.  Further research is still warranted.

Results

Line 224.  This line is rather awkwardly placed at the bottom of the page.  Can this be reformatted more readably?

Line 234.  Table 2 should be moved up to here.

Line 240.  Table 3 should be moved up to here.

Table 2. Based on the Directed Acyclic Graph to evaluate the effect of potential confounders and the authors’ thoughtful analysis thereof, they chose a very conservative final model.  Based on the interaction between agricultural work, height, and smoking with both the independent and dependent variables, there appears to be a strong case to include these in the final model.  At a minimum, I suggest including the Fully adjusted (based on DAG) model in Table 2, or possibly use that model as the final model for the entire study.  This would not be unreasonable.

Figure S4: Generalised additive model of the association between E-DII and CLL (p= 0.18)

This figure adds significantly to the results and deserves a place in the manuscript, not just the Supplement.

Author Response

Reviewer #3:

Comments and Suggestions for Authors

This is a well-conducted case control study of chronic lymphocytic leukemia in Spain.  The well thought out logistic analysis showed no statistical association with the Dietary Inflammatory Index.  Here are a few suggestions for changes.

Abstract

Line 74.  I suggest adding the word “significantly” before “associated”.  This will stress that while the DII-CLL association did not meet the p<0.05 threshold, there was a clear trend in that direction.

As suggested by Reviewer 3, we have added “significantly” in the conclusion of the abstract:

“In conclusion, consumption of a diet with high pro-inflammatory components was not significantly associated with CLL.”

Line 75. I suggest removing the word “Dramatic” here.  Given the modest findings of this study, it is unlikely that dramatic findings in a younger population can be expected.  Further research is still warranted.

As suggested by Reviewer 3, we deleted the word “dramatic” in the conclusion of the abstract.

Results

Line 224.  This line is rather awkwardly placed at the bottom of the page.  Can this be reformatted more readably?

As suggested by Reviewer 3, we moved slightly down the table 1 to avoid splitting into two parts the text.

Line 234.  Table 2 should be moved up to here.

Following the suggestion from Reviewer 3, Table 2 is now included in section “3.2. E-DII and CLL risk” to ease the reading of the manuscript.

Table 2. Based on the Directed Acyclic Graph to evaluate the effect of potential confounders and the authors’ thoughtful analysis thereof, they chose a very conservative final model.  Based on the interaction between agricultural work, height, and smoking with both the independent and dependent variables, there appears to be a strong case to include these in the final model.  At a minimum, I suggest including the Fully adjusted (based on DAG) model in Table 2, or possibly use that model as the final model for the entire study.  This would not be unreasonable.

We thank the reviewer for this thoughtful comment. We added the fully adjusted model based on the DAG in Table 2.

Figure S4: Generalised additive model of the association between E-DII and CLL (p= 0.18)

This figure adds significantly to the results and deserves a place in the manuscript, not just the Supplement.

The Figure S4 is now included in the main manuscript and has been labelled “Figure 2”. Following this change, the supplementary material Table S5 and List S6 have been labelled as Table S4 and List S5.

Reviewer 4 Report

Here are the comments for the manuscript id: 638270

In the present work by Jose Carlos Flores et al,. authors tried to find out co-relation between dietary inflammatory index (DII) and CLL incidences and they observed modest, non-significant correlation between DII and CLL.
Authors themselves have mentioned tha the lack of power which could be on of the reasons to have no significant correlation in any of the parameters tested.
More description of comorbid condition is required in this study.
Cigarette smoking is known to be a proinflammatory agent and has been associated with many chronic inflammatory dieseases and in cancer as well. It is highly surprising to see the increased CLL in non-smokers.
This manuscript lacks complete data and most of the data are not shown.

Manuscript needs to redone and experiments are not done correctly.

Round 2

Reviewer 4 Report

Manuscript is improved now and reviewer has no further comments.
They added some information in the main manuscript.